# Trees and their seed networks: The social dynamics of urban fruit trees and implications for genetic diversity

**Aurore Rimlinger**[1,2]*, **Marie-Louise Avana**[3], **Abdon Awono**[4], **Armel Chakocha**[3], **Alexis Gakwavu**[3], **Taïna Lemoine**[5], **Lison Marie**[2], **Franca Mboujda**[2,3], **Yves Vigouroux**[2], **Vincent Johnson**[6], **Barbara Vinceti**[6], **Stéphanie M. Carrière**[1☯]*, **Jérôme Duminil**[2,6☯]*

**1** SENS, IRD, CIRAD, Univ. Paul Valery Montpellier 3, Univ. Montpellier, Montpellier, France, **2** DIADE, Univ. Montpellier, IRD, Montpellier, France, **3** Forestry Department, Faculty of Agronomy and Agricultural Sciences, University of Dschang, Dschang, Cameroon, **4** CIFOR, C/o IITA Humid Forest Ecoregional Center, Yaoundé, Cameroon, **5** Université de Montpellier, Montpellier, France, **6** The Alliance of Bioversity International and CIAT, Fiumicino Rome, Italy

☯ These authors contributed equally to this work.
* aurore.rimlinger@ird.fr (AR); stephanie.carriere@ird.fr (SMC); jerome.duminil@ird.fr (JD)

**Data Availability Statement:** Databases and scripts used to (i) calculate and compare seed provenances between the urban and rural populations; (ii) identify the main departments of origin of African plums sold in Yaoundé; (iii)

## Abstract

Trees are a traditional component of urban spaces where they provide ecosystem services critical to urban wellbeing. In the Tropics, urban trees' seed origins have rarely been characterized. Yet, understanding the social dynamics linked to tree planting is critical given their influence on the distribution of associated genetic diversity. This study examines elements of these dynamics (seed exchange networks) in an emblematic indigenous fruit tree species from Central Africa, the African plum tree (*Dacryodes edulis*, Burseraceae), within the urban context of Yaoundé. We further evaluate the consequences of these social dynamics on the distribution of the genetic diversity of the species in the city. Urban trees were planted predominantly using seeds sourced from outside the city, resulting in a level of genetic diversity as high in Yaoundé as in a whole region of production of the species. Debating the different drivers that foster the genetic diversity in planted urban trees, the study argues that cities and urban dwellers can unconsciously act as effective guardians of indigenous tree genetic diversity.

## Introduction

Crop seed exchange networks, shaped by social dynamics, have a deep influence on the organization and breadth of plant diversity in human-managed environments. The decisive effect of social organizations, through bonds of kinship, marriage or friendships, has been shown to influence the flows of crop planting materials [1–3]. In turn, crop species' diversity within rural home gardens is influenced by exchanges of planting material (seeds or clonal material) [4, 5]. In urban environments, although the propagation of plants by humans has been described through the lens of their accidental role in propagule dispersal [6, 7], there are only few studies mentioning intentional seed circulation patterns for crop species, and notably for perennial crop species such as fruit trees. In South America, it was shown that seeds and

calculate the genetic parameters of the urban and rural sampled; were deposited in the Open Science Framework database (https://osf.io/6vjny/).

**Funding:** This project has been supported by Agropolis Fondation (https://www.agropolis-fondation.fr/) under the reference ID 1605-042 awarded to J.D. through the «Investissements d'avenir» program (Labex Agro: ANR-10-LABX-0001-01), under the frame of I-SITE MUSE (ANR-16-IDEX-0006). We acknowledge the Agence Universitaire de la Francophonie (AUF) for funding the data collection. The funders had no role in study design, data collection and analysis, decision to publish, or preparation of the manuscript.

**Competing interests:** The authors have declared that no competing interests exist.

seedlings from urban home gardens were acquired at local markets, but also through networks of exchanges involving relatives and neighbours [8, 9]. Together with the major contribution of natural dispersion and historical plantings in shaping the plant diversity present in urban home gardens, the ones of local nurseries and social networks were also emphasized in the San Juan area, in Puerto Rico [10]. In Amazonia, the contribution of rural genetic material to urban gardens has also been demonstrated: the major source of exchanged planting material from urban home gardens is through gifts, which come both from communities and households in the city, and from those in rural areas [11]. Conversely, the role of urban centres as sources of planting material has also been exemplified, with rural gardeners purchasing planting material closer to urban centres [12, 13]. These different findings highlight the need to understand the extent to which rural-urban bonds may be drivers of crop plant intra- and inter-specific diversity in urban environments.

Trees are a traditional component of urban spaces, mostly planted for ornamental purposes or for their shade. The multi-faceted role of urban trees is now well recognized since they provide ecosystem services critical to urban citizen wellbeing: green places for leisure, noise reduction, climate mitigation, air and water purification, energy savings, habitats for biodiversity, and carbon sequestration [14–16]. In tropical regions, urban trees also contribute to diets [17, 18]. In Kinshasa, fruit production was cited as the main reason why urban dwellers introduced trees around their home, with trees providing additional services (shade, medicine, cash), such as African plum trees, avocado trees and mango trees, being highly valued [19]. Among the studies that have characterized urban food-tree species diversity and composition within cities in Central Africa [20–24], the origin of planting material is sometimes questioned by separating local (native) and exotic (non-native) trees. But the information on the source of the planting material for local trees is not available. This question matters all the more since the social dynamics of tree planting influence the distribution of tree genetic diversity.

Genetic diversity is the raw material for species adaptation to global changes such as new climates, pests or diseases, or pollution. This fundamental species survival mechanism, that has taken millions of years to develop, is critically threatened by anthropogenic disturbances such as overexploitation, deforestation, land-use change, or climate change. On the other hand, humans can play a positive role in the conservation of species' genetic resources. In particular, research shows that through their traditional practices and local ecological knowledge, tropical smallholder farmers actively participate in safeguarding agricultural biodiversity [25–28]. By maintaining many different crops and varieties, farmers retain a range of options for adapting to environmental change and thus promoting resilience in agricultural systems [29, 30].

There have been very few studies on the social dynamics of urban fruits trees planting and their influence on species genetic diversity. Our study explored how urban intra-specific diversity can be embedded within human social life [31]. We developed a multi-disciplinary framework bringing together ethnoecological and genetic data to (i) assess the sources used by urban dwellers to plant new trees; and (ii) evaluate the consequences on the urban intra-specific genetic diversity. Our study examined the seed origins and genetic diversity of an urban population of a major indigenous fruit tree species from Central Africa, the African plum tree (*Dacryodes edulis*, (G. Don) H.J. Lam, Burseraceae), and underlined its specificity by contrasting it with those of a larger rural population.

## Materials and methods

### Species description

*Dacryodes edulis* (G. Don) H.J. Lam (Burseraceae) is a culturally important fruit tree species originating from the Congo Basin. It is mostly cultivated as a fruit and shade tree in coffee-

cocoa agroforests [32], and in home gardens (see pictures in S1 Fig). *Dacryodes edulis* is a pre-dominantly outcrossing species [33]. It is pollinated by insects, especially Apoidea [34]. In natural populations, apes, birds and mammals act as seed dispersers, but anthropogenic seed dispersal is more important in cultivated populations, as seeds and fruits are commonly exchanged between relatives, neighbours, and villages [35]. The fruits (African plums) are one-seeded drupes. Fruits present a wide range of sizes, shapes, colours (epicarp, mesocarp) and textures [36, 37]. They contain more than 50% lipids, but also proteins, fibre and vitamins [38, 39]. Their fleshy, buttery pulp is highly popular, and is consumed with roasted maize, plantain, or tubers. African plums are part of the diet of all cultural groups in tropical Cameroon.

## Ethnoecology data collection and analysis

To investigate the social dynamics of urban tree planting, we selected the city of Yaoundé (3.9 million inhabitants), the second largest city in Cameroon. The founding of the city dates back to the late 19th century. Along with most of the large sub-Saharan cities, it has quickly expanded and stretches now over more than 300 km$^2$. Agricultural activities are carried out both in its urban core and in the surrounding peri-urban areas, ensuring cheap food supplies. Yaoundé's cultural mosaic reflects its urbanization history: because of the destruction and resettlement of neighbourhoods during the colonial era, the city's districts comprise mixed populations coming from different cultural and geographical areas of Cameroon [40, 41]. *Dacryodes edulis* trees are present throughout the city, especially on its outskirts where the home gardens are mostly bigger, and in low-income neighbourhoods where home gardens contain more useful plant species [42].

We led interviews with tree owners (N = 84) in one neighbourhood of Yaoundé, Oyom-Abang, and recorded their cultural origins. This neighbourhood is culturally diverse (different cultural groups are represented, in order of importance in our sampling: Beti, Bassa and Bamileke) and presents a moderate to dense urbanization level, with approximately 105 inhabitants per hectare. Despite urbanization, a network of trees and gardens is interspersed among buildings, with many fruit trees present in backyard home gardens. These home gardens are generally small, located around the family compound, and include mixed crops (fruits, vegetables, tubers). The selection of respondents was based on their ownership of one or more *D. edulis* trees. Prior to the interview, tree owners were informed of the research intentions and of their right to participate or decline. At the end of the interview, tree owners were given a form stating that the interview had been conducted in accordance with the principles of free and informed consent, which they could sign if they agreed.

For comparison with the urban area, the African plum tree planting practices were also recorded with tree owners (N = 47) in a rural area. The rural area is located in the West region, one main production area of *D. edulis* fruits in Cameroon [43], which supplies the major markets of the two biggest cities in Cameroon, Yaoundé and Douala. Different cultural groups are present throughout the region, the most numerous being Bamileke and Bamoun. Cultivators of this region focus on the production of cash crops, especially cocoa and coffee, but integrate many other fruit trees species in their fields.

We obtained information on the location from where the propagation material (seeds, seedlings) originated for 121 *D. edulis* trees in the urban area. In the rural area, this information was recorded, for trees present in home gardens and fields, in two villages, Manjo and Bandounga (94 trees in total). Four different categories of seed origin were distinguished: the seeds could have been taken directly by the farmer from his own field/home garden (category *farmer's own trees*); the seeds could have been taken outside of the farmer's own field/home garden but still originate from either the same village (category *same village*) or from a different village

(category *different village*); or the seeds could have been bought at the market or in a nursery (category *market or nursery*), meaning that the fruit has travelled through a commercial exchange network and possibly originates from far away. For the urban population, the seeds could come from the village of origin of urban dwellers with a rural background, which was made explicit with the subcategory *% coming from the village of origin* in the category *different village*.

Given the high proportion of seeds sown from fruits bought in markets in Yaoundé, the origin of the fruits sold on these markets was traced back through interviews with a total of 89 sellers of African plums, in nine markets of Yaoundé (Emana, Etoudi, Marché des fruits, Mendong, Mfoundi, Mokolo, Mvog Mbi, Nkolbisson, Nsam) from July to September 2017.

## Genetic data collection and analysis

To understand the link between urban tree planting behaviours and species genetic diversity characteristics in the city we analysed the level of genetic diversity from the urban population of Oyom-Abang (250 ha) and compared it to that of one rural population stretching over a main production area of *D. edulis* fruits in Cameroon (200,000 ha, Fig 1). In this aim, we sampled 450 trees in Oyom-Abang (164 trees included in the ethnoecological dataset + 286 additional trees sampled only for the genetic characterization) and 399 trees in the rural population

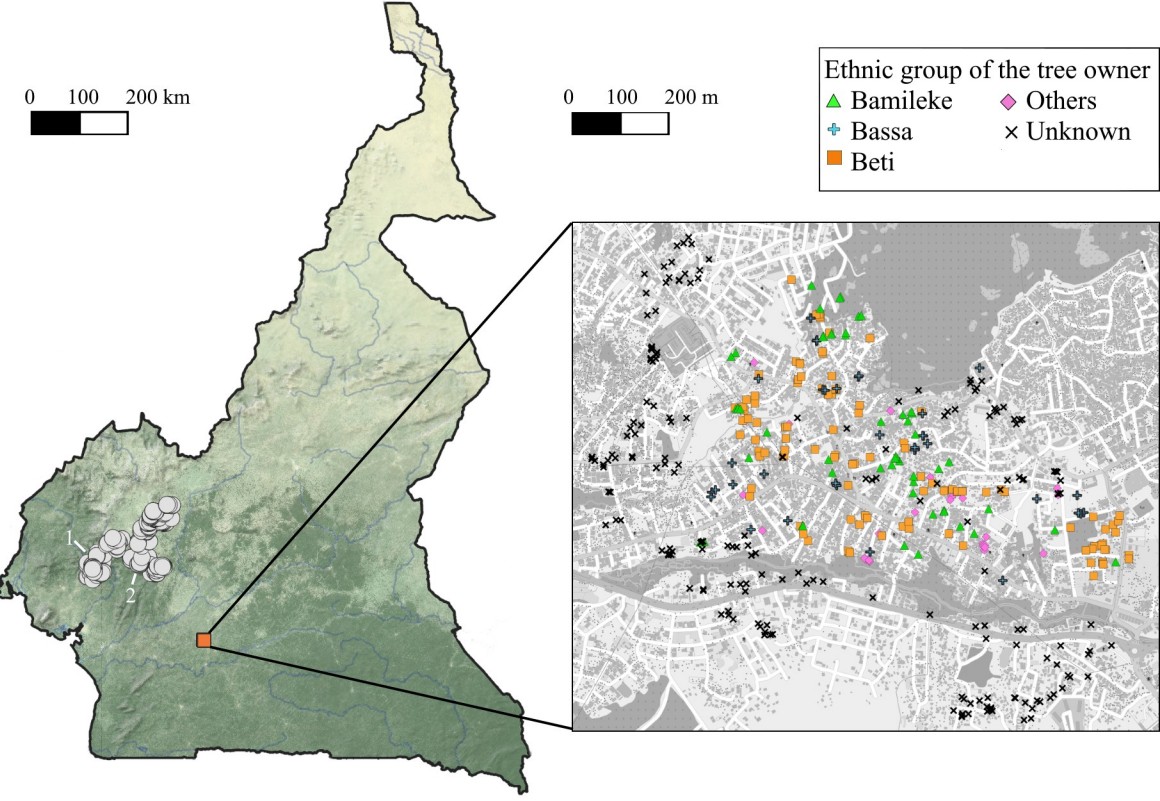

**Fig 1. Map of Cameroon showing the two sample populations.** The large rural population on the left (white dots) comprises the two localities where the social dynamic of tree planting was investigated (1, Manjo; 2, Bandounga). The urban population (Yaoundé, orange square) is enlarged on the right, showing the urban trees sampled in the 250-ha area, with information on the ethnic group of the tree owners. Map image was created on QGIS version 3.6.1 [44]. We used OpenStreetMap (www.openstreetmap.org) and shapefiles of Earth with shaded relief and water from Natural Earth (www.naturalearthdata.com/ and of the global tree cover in the area, described in Hansen *et al.* [45] and available at glad.umd.edu/Potapov/TCC_2010/. GPS coordinates are stored in the S1 Dataset.

(94 trees from the ethnoecological dataset + 305 additional trees), that stretches across the West, Littoral and Centre regions (278, 79 and 42 trees respectively, in a total of fifteen villages).

We extracted the DNA of these 849 urban and rural samples following the protocol of Mariac *et al.* [46] (S1 Dataset). The genetic diversity was investigated using fourteen nuclear microsatellite markers amplified in two multiplexes following Rimlinger *et al.* [47]. All the individuals were genotyped using an ABI 3500 XL sequencer (Applied Biosystem, Foster City, California, USA) at the CIRAD Genotyping Platform in Montpellier, France. Electropherograms were visualized and scored with the microsatellite plugin in Geneious 7.1.3 (https://www.geneious.com). For each locus and population, observed and expected heterozygosity ($H_O$ and $H_E$), inbreeding coefficient ($F_{IS}$), the effective number of alleles (Ne), the rarefied allelic richness (AR), null allele frequency (r) and the corrected inbreeding coefficient corrected for the presence of null allele ($F_{null}$) were estimated using INEst 2.2 [48] and SPAGeDi [49]. The level of population differentiation (as estimated by the $F_{ST}$ fixation index) and the test of genetic structure (permutation of individuals among all populations) were obtained with SPAGeDi. Levels of diversity between the urban and rural populations were compared with one-way analysis of variance, controlling for the loci effect. The distribution of the genetic diversity between the two populations was further tested using the Bayesian clustering analysis implemented in STRUCTURE [50] with admixture ancestry, correlated allele frequencies and no prior information about population origin. K was set from 1 to 10, and each run was replicated 10 times, with a burn-in period of 10,000 followed by 40,000 Markov Chain Monte Carlo repetitions. To visualize the distribution of genotypes of the rural and urban populations, a principal component analysis was performed on the microsatellite dataset using the R package adegenet [51].

## Results

From the interviews conducted with tree owners on their private land, the geographic origin of the seeds used for plantation indicated wide differences between the rural and urban tree populations (Table 1).

In Yaoundé, 93% of *D. edulis* planted seeds came from beyond the city borders, with half of the seeds used for planting coming from markets, either from Yaoundé markets or from more distant locations known for the quality of their African plums (Makénéné's market for instance, located 200 kms away from Yaoundé). Fruits sold on Yaoundé's markets come from five main areas, located at various distances from the capital, up to 370 km away (S2 Fig). The second most important source of seed is provided by rural villages beyond Yaoundé's border (36%). The main cultural groups present in the sampled urban population are Bamileke, Bassa and Beti (Fig 1), who originate from different regions (respectively western Cameroon, the Littoral and the central region, an area partly covered by this study, see the Methods section).

**Table 1. Differences in seed provenance and distance between rural and urban populations of *D. edulis*.**

| Population | | Seed provenance (frequency) * | | | | Seed distance (km) * |
|---|---|---|---|---|---|---|
| | N | Farmer's own trees | Same village | Different village (% village of origin) | Market or nursery | All provenances combined |
| **Urban** | 121 | 0.04 | 0.03 | 0.36 (81%) | 0.57 | 45.4 ± 5.9 |
| **Rural** | 94 | 0.30 | 0.46 | 0.06 | 0.18 | 3.7 ±0.8 |

* the seed provenance and seed distance between the two populations are significantly different (as tested respectively with the Pearson's Chi-squared test and Wilcoxon rank sum test; p-value < 2.2e-16).

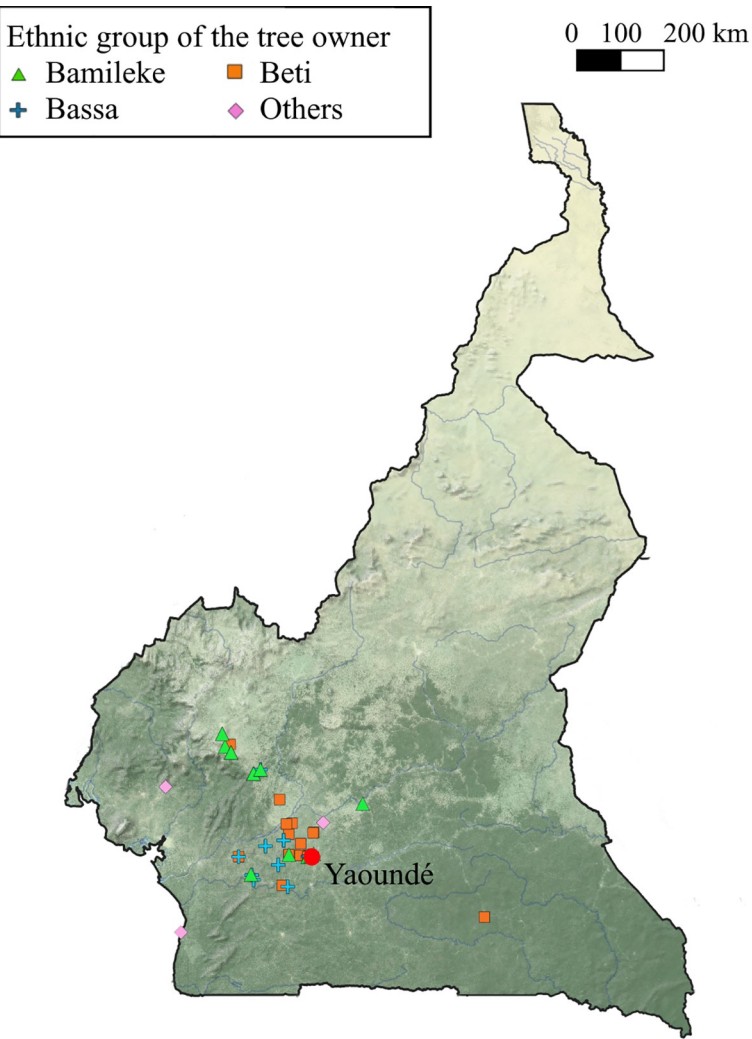

**Fig 2. Provenances of the seeds used to plant different urban trees in Yaoundé.** The different trees sampled in Yaoundé (n = 121, red dot) are mapped with symbols depending on the ethnic group of the tree owners. Map image was created using the same program and shapefiles as Fig 1. GPS coordinates indicating seed provenance can be found on the OSF repository of the project.

Travelling back to their villages is an opportunity for urbanites to bring back fruits and seeds, and the diversity of geographical origins of the cultural groups settled in the city has led to planting seeds from different locations (Fig 2). In contrast, in the rural population, more than 75% of the trees came from the same village territory. These differences of origins translate into wide-ranging differences (Table 1, Wilcoxon signed rank test, p-value < 2.2e-16, effect size = 0.514) in the average distance travelled by seeds.

The level of genetic diversity (rarefied allelic richness for k = 300 gene copies) was high and similar (p-value = 0.945, Wilcoxon rank sum test) in the urban and in the rural populations despite large differences in the geographical coverage (250 ha for the urban population, 200,000 ha for the rural one) (Table 2). The genetic composition is largely similar between the two populations as shown by (i) the weak but significant genetic differentiation ($F_{ST}$ = 0.0057

Table 2. Genetic diversity indices of the rural and urban populations of *D. edulis*.

| | N | NA | NAe | AR | $H_E$ | $H_O$ | $F_{IS}$ | $F_{null}$ |
|---|---|---|---|---|---|---|---|---|
| **Urban population** | 450 | 18.3 ± 2.9 | 6.17 ± 1.1 | 15.7 ± 2.5 | 0.761 ± 0.039 | 0.629 ± 0.051 | 0.181 ± 0.047 | 0.0201 ± 0.0012 |
| **Rural population** | 399 | 18.8 ± 2.7 | 6.40 ± 1.2 | 16.1 ± 2.3 | 0.768 ± 0.038 | 0.659 ± 0.045 | 0.149 ± 0.036 | 0.018 ± 0.0014 |

NA: number of alleles; Ne: effective number of alleles, AR: rarefied allelic richness for k = 300 gene copies; $H_E$: expected heterozygosity; $H_O$: observed heterozygosity; $F_{IS}$: inbreeding coefficient; $F_{null}$: inbreeding coefficient corrected for null alleles; values are means ± SEM.

and is significantly different from 0 as measured by an exact G test, p-value <2.2e-16); (ii) the principal component analysis (S3 Fig); (iii) the Bayesian clustering analysis where individuals are attributed to two (weakly differentiated) genetic clusters without relation to their population of origin (S4 Fig).

## Discussion

Urban contexts embody a complex socio-cultural nexus where food supply to humans is crucial [52]. As shown by this study, similar levels of genetic diversity were observed between the urban and rural tree populations despite large differences in the geographical coverage. The planting practices of urban trees with regards to cultivated fruit trees in the cities can thus generate positive evolutionary dynamics for their diversity. This can be explained by two intertwined processes.

First, big cities are at the heart of market flows and social networks. A vast majority of food produced in rural areas reach cities to be sold in marketplaces. If particularly appreciated, the seed from a good fruit is preserved and planted within the family compound, hence safeguarding its genetic material. This process of fruit exchanges through commercialization from production/rural areas to large cities may account for one of the most important drivers of urban dynamics in favour of the genetic diversity of native fruit trees, whereby most genetic resources of a given valuable fruit species are potentially channelled to cities, being 'attracted' by consumers' demand. Diversity in an urban setting tends to be high not only because there is a lot of inflow of plant material from different surrounding regions but also because seeds used for planting harbour a significant proportion of the genetic diversity of their population of origin given the predominantly outcrossing mating system of the species [33].

Second, this large exchange matrix of genetic material is further enhanced by what is known as an 'economy of affection' matrix [53], which refers to the strong affective links shared within a group of people with the same socio-cultural background. For people who migrated to an urban context, this bond includes a relationship with nature and the crops (from species to varieties) from the rural area of origin. Through gift-giving chains, urban dwellers can also have access to wild plants from rural areas [54]. Urban dwellers thus commonly commented that "the best fruits [were] those cultivated in the village of my family". This fits with the more general influence of kinship systems on farmers' seed exchange networks [1, 55, 56]. As big cities attract people from all over the surrounding localities and from various cultural groups, this may be another important factor that determines a concentration of genetic resources from different rural areas in cities [57].

The way tree planting material is sourced in cities is therefore critical for the current pattern of diversity. The low levels of tree genetic diversity reported in some cities of temperate regions was explained by their reliance on poorly diversified planting material (selected and clonal material) from commercial nurseries [58]. The genetic diversity observed in Yaoundé could thus be rooted in the informal seed exchange system, where seeds are loosely selected and planted by urban dwellers.

The diversity of *D. edulis* trees planted in gardens is supported through exchange networks that include kin, friends, and outsiders. The strong reliance on local systems of production and exchange, and the maintenance of rural-urban links, fuel germplasm transfers and lead to unintended positive consequences, such as conservation outcomes and urban sustainability [11]. According to traditional conservation perspectives, anthropogenic actions are framed as a major cause of biodiversity erosion [59], or as being deliberately designed to safeguard targeted species and ecosystems [60]. However, there is a third way where the sum of individual actions contributes to biodiversity conservation: when pursuing other goals, here coping with subsistence needs and maintaining socio-cultural cohesion, they may also maintain genetic diversity of useful crop species [61]. A broader analysis replicating the framework of this study for other crops [62] and cities pantropically would show how important this serendipitous "safeguarding" of genetic diversity is. Urban home gardens can actually contain a significant proportion of indigenous species (35–60%) which represent additional candidates to test the importance of urban areas for the conservation of species genetic diversity [63, 64].

According to the United Nations [65], more than two-thirds of the world will live in urban areas by 2050, having major direct and indirect impacts on biodiversity [66]. It is therefore essential to give to the biodiversity of these anthropogenic landscapes the special attention they deserve [67] as well as to promote the contribution of urban trees to human food supply and other ecosystem services. However, with the anarchic urbanization in the rapidly expanding cities of developing countries, these trees are at risk of being wiped out. There is thus an urgent need to identify the main socioeconomic, cultural and political drivers as well as more effective urban biodiversity conservation instruments which may contribute to conserve and further diversify fruit trees in the city [68–70]. A focus on genetic diversity conservation, promoted by socially desirable measures, can help ensure green, healthy, food provisioning and more resilient cities sheltering crop plant species with a high adaptive potential, in the face of climate change and growing urban populations [71, 72]. Concomitantly with the conservatories and botanical gardens, parks, orchards and individual backyards, which actively support the goal of conserving biodiversity, it is time to better understand, analyse and promote urban home gardens and green spaces as repositories of tree genetic diversity.

## Supporting information

**S1 Fig. Pictures of the species.** Mature tree in an agroforest (above left); young tree in an orchard near Koutaba (above right). Roasted African plums and plantains sold in the streets (below left); fruit varietal diversity (below right). Pictures from the authors (1,2,4: A. Rimlinger; 3: J. Duminil).
(PDF)

**S2 Fig. Provenance of fruits sold in Yaoundé markets.** Provenance of African plums in Yaoundé main markets, accounting for more than 95% of the total volume of sold fruits reported by the sellers (1: Ndé; 2: Moungo; 3: Nkam; 4: Mbam-et-Inoubou; 5: Lékié; 6: Nyong-et-Kéllé). Dots correspond to the rural trees sampled for the genetic analysis. Administrative boundaries of Cameroon were added using a shapefile available at data.humdata.org/dataset/cameroon-administrative-boundaries. The source of the map data are stored in a database (market_provenance.txt) deposited in the OSF repository of the project.
(PDF)

**S3 Fig. Principal component analysis (PCA) of Dacryodes edulis microsatellite diversity from the urban and rural populations.** The PCA was performed on the allele frequencies of each individual. Genotypes were clustered to show maximal differentiation along the first and

second principal component (PC1 and PC2). Individuals do not cluster according to their urban or rural origin, indicating the absence of population structure.
(PDF)

**S4 Fig. Bayesian clustering analysis. S4a**: Changes in K values from the mean log-likelihood probabilities (right axis) and plot of mean likelihood L(K) and variance per K value (left axis) from STRUCTURE runs where inferred clusters (K) ranged from 1 to 10. **S4b:** Output of clustering analysis by STRUCTURE software for two clusters (K = 2) of the 849 trees, using fourteen microsatellite markers, grouped by origin (urban trees, rural trees). Each vertical bar represents one individual and shows its inferred cluster membership; black and gray colors correspond each to one cluster. If both colors are present, the haplotype consists of a mixture of markers assigned to both black and gray clusters. The samples from the urban area and rural area were assigned in different proportions to each cluster. Using an assignment probability threshold of 0.8, 41% and 8% of individuals from the urban population were respectively assigned to the black and gray clusters (51% were presenting intermediate genotypes) and 5% and 45% of individuals from the rural population were respectively assigned to the black and grey clusters (50% were presenting intermediate genotypes).
(PDF)

**S5 Fig.**
(JPG)

**S1 Dataset.**
(XLSX)

## Acknowledgments

The authors thank Aurélien Nguegang and Saïd Fewou Njoya for assisting in the sample collection, and Christian Leclerc and Adeline Barnaud for helpful comments on the first draft version.

## Author Contributions

**Conceptualization:** Marie-Louise Avana, Abdon Awono, Yves Vigouroux, Stéphanie M. Carrière, Jérôme Duminil.

**Formal analysis:** Aurore Rimlinger, Lison Marie.

**Funding acquisition:** Jérôme Duminil.

**Investigation:** Aurore Rimlinger, Armel Chakocha, Alexis Gakwavu, Taïna Lemoine, Lison Marie, Franca Mboujda.

**Methodology:** Aurore Rimlinger, Abdon Awono, Stéphanie M. Carrière, Jérôme Duminil.

**Project administration:** Jérôme Duminil.

**Resources:** Jérôme Duminil.

**Supervision:** Marie-Louise Avana, Abdon Awono, Stéphanie M. Carrière, Jérôme Duminil.

**Validation:** Aurore Rimlinger, Jérôme Duminil.

**Writing – original draft:** Aurore Rimlinger, Stéphanie M. Carrière, Jérôme Duminil.

**Writing – review & editing:** Aurore Rimlinger, Yves Vigouroux, Vincent Johnson, Barbara Vinceti, Jérôme Duminil.

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
