## [Decision Letter · Decision Letter 0]

29 Jan 2021

PONE-D-20-35472

Trees and their seed networks: the social dynamics of urban fruit trees and implications for genetic diversity

PLOS ONE

Dear authors,

Thank you for submitting your manuscript to PLOS ONE. After careful consideration, we feel that it has merit but does not fully meet PLOS ONE’s publication criteria as it currently stands. Therefore, we invite you to submit a revised version of the manuscript that addresses the points raised during the review process.

We look forward to receiving your revised manuscript.

Kind regards,

Ji-Zhong Wan

Academic Editor

PLOS ONE

Journal Requirements:

2. We note that Figures 1 and 2 and S2 Figure in your submission contain map and satellite images which may be copyrighted.

a. You may seek permission from the original copyright holder of Figures 1 and 2 and S2 Figure to publish the content specifically under the CC BY 4.0 license. 

Additional Editor Comments:

This is an interesting study. However, the authors have to address all the concerns of the three reviewers before publication.

Reviewers' comments:

Reviewer's Responses to Questions

**Comments to the Author**

1. Is the manuscript technically sound, and do the data support the conclusions?

Reviewer #1: Yes

Reviewer #2: Partly

Reviewer #3: Yes

2. Has the statistical analysis been performed appropriately and rigorously? 

Reviewer #1: Yes

Reviewer #2: Yes

Reviewer #3: Yes

3. Have the authors made all data underlying the findings in their manuscript fully available?

Reviewer #1: No

Reviewer #2: Yes

Reviewer #3: Yes

4. Is the manuscript presented in an intelligible fashion and written in standard English?

Reviewer #1: Yes

Reviewer #2: Yes

Reviewer #3: Yes

5. Review Comments to the Author

Reviewer #1: This work is a case study on the impact of human activities on the levels and patterns of genetic diversity of an indigenous fruit tree in Yaoundé, Central Africa. It’s seems interesting to me, and both the experiment design and data analyses of this work are clear, logic and reliable. However, genetic diversities of urban trees are influenced by many factors, such as the economic values, availability of wild species, human activities, culture, seeds sources used for seedlings, et al. So, the implication of this study may be limited and depend on the specific tree species used in studying.

1. There are no figure legends for Fig. 1 and Fig. 2.

2. There is no description on the Principal component analysis. How does this analysis done?

3. Line 202, “the weak but significant genetic differentiation, FST = 0.0057, p < 2.2e-16”. The value of FST is quite small, indicating there is no apparent differentiation between urban and rural populations.

4. Line 165 and Line 260, Table 1, use Ne for “effective number of allele”

5. How many microsatellite markers are indeed used? In Line 158, the author said that 12 SSR were used in this study, but in the figure legend of S4b Figure, at Line 9, it said “using 14 microsatellite markers”.

6. The D4_fig is not clear. Please improve the its resolution.

7. In the figure legend of S4b Figure, Line 13-17, the urban and rural populations differed significantly in the proportion of the two genetic clusters. This seems to be incompatible with the PCA result and the estimated population differentiation (FST = 0.0057). The authors should check the result of STRUCTURE analysis and make sure the result of STRUCTURE is reasonable or not.

Reviewer #2: ‘Trees and their seed networks: the social dynamics of urban fruit trees and implications for genetic diversity’ is an interesting study exploring the genetic diversity present in the African plum tree (Dacryodes edulis) in urban areas. The manuscript is well written, follows a clear logic flow and describes an important aspect of our biodiversity.

My main critical point is around the genetic analysis. As these are not natural populations most Hardy-Weinberg assumptions (used for HE, Ho, Fis, Fst) will be violated, i.e. having a closed population. In many cases when working with wild populations HWE assumptions will be slightly violated but in this case seeds / seedlings are coming from all over the place and the trees analysed will most likely never have formed a natural population.

More specific comments:

- Material and Methods, Species description: Could you include information about pollinators and seed dispersers, i.e. is gene flow likely to be predominantly long- or short-distance

- Results: You’re describing the genetic diversity as ‘high’ but high compared to what? Natural populations?

- Results: When you say your STRUCTURE analysis clustered your samples into two ‘weakly’ differentiated clusters, what does that mean? STRUCTURE doesn’t usually give strong or weak clusters.

- Results: Table 2: As I understand you collected samples from more than 2 populations but averaged and split into 2 populations (Urban and Rural)? If so, was there a difference in AR between the individual populations?

- Results: As explained above I don’t think you can do HWE based population genetic analysis for this study as the results will be strongly affected by the violation of the assumptions. You can still look at AR and use the STRUCTURE analysis but not HE, HO, FIS, FST. Unless some of your sampled populations are indeed natural and not planted?

- Results: Please include on how many individuals AR is based on?

- Results: It would be good to have a bit more information about, i.e. how many alleles did you find in total, did all samples work, were markers tested for things like linkage or null alleles

- Discussion: It would be good if you could demonstrate that the samples you’ve analysed are indeed genetically diverse. If the species has been analysed genetically in natural populations with the same marker set you could such a comparison. Without that it’s not really possible to say whether diversity is actually high or low in urban areas. It’s possible that it’s high but it might also be significantly lower compared to other areas. If no such studies exist, then you should at least discuss this in the discussion.

Reviewer #3: Dear Editor and Authors,

The manuscript is well-written with valuable insights on the genetic diversity of an urban African plum tree in Central Africa and their extended network.

From my point of view, findings from this study are of importance significance to be published to scientific community, especially for population diversity researches.

However, figures provided are lack of quality, which I think might affect the manuscript's output.

I recommend minor revision for this manuscript, only after figures are replaced with high-quality images.

6. PLOS authors have the option to publish the peer review history of their article (what does this mean?). If published, this will include your full peer review and any attached files.

Reviewer #1: No

Reviewer #2: No

Reviewer #3: **Yes: **Dzarifah Zulperi

---

## [Author Response · Author response to Decision Letter 0]

15 Feb 2021

We addressed the three additional requirements (style requirements, replacement of copyrighted map figures, and inclusion of figure captions). The details are provided in our Response to the reviewers.

---

## [Editor Report · Decision Letter 1]

17 Feb 2021

Trees and their seed networks: the social dynamics of urban fruit trees and implications for genetic diversity

PONE-D-20-35472R1

Dear Dr. Rimlinger,

We’re pleased to inform you that your manuscript has been judged scientifically suitable for publication and will be formally accepted for publication once it meets all outstanding technical requirements.

Kind regards,

Ji-Zhong Wan

Academic Editor

PLOS ONE

Additional Editor Comments (optional):

All the comments have been addressed. I believe that it is a small case on trees, but very interesting study.
---

## [Editor Report · Acceptance letter]

3 Mar 2021

PONE-D-20-35472R1 

Trees and their seed networks: the social dynamics of urban fruit trees and implications for genetic diversity 

Dear Dr. Rimlinger:

I'm pleased to inform you that your manuscript has been deemed suitable for publication in PLOS ONE. Congratulations! Your manuscript is now with our production department. 

Kind regards, 

on behalf of

Dr. Ji-Zhong Wan 

Academic Editor

PLOS ONE